# Inclusive Play: Defining Elements of Playful Teaching and Learning in Culturally and Linguistically Diverse ECEC

Jonna Kangas [1,2,*], Anna-Leena Lastikka [1] and Outi Arvola [2,3]

1   Faculty of Educational Science, University of Helsinki, 00014 Helsinki, Finland; anna-leena.lastikka@helsinki.fi
2   Kokkola University Consortium Chydenius, University of Jyväskylä, 67701 Kokkola, Finland; outi.a.m.arvola@jyu.fi
3   Faculty of Education, University of Turku, 20014 Turku, Finland
*   Correspondence: jonna.kangas@helsinki.fi

**Abstract:** Inclusive values are integral to early childhood education and care (ECEC) policies, promoting equal participation opportunities and individual support for all children. Play serves as a method for meaningful engagement, mutual cultural knowledge creation, and learning within ECEC. Pedagogical play entails teachers' observation, interaction, support, and guidance. This study investigates practical elements and methods employed by teaching staff and experienced by children during playful activities. Conducted as a case study in two culturally diverse ECEC centers during spring 2023, data collection involved video-recorded monitoring of children's daily activities in five groups. Video data were transcribed and analyzed using content analysis tools to identify categories of inclusive play. Findings are presented as narratives to honor children's experiences. The study identifies five elements of inclusive play: teachers' active participation and presence, balanced repetition with flexible plans and adaptive goals, playful language fostering joy in play, non-verbal and kinesthetic communication with enabling tools, and emerging play marked by interaction and lasting intensity. These elements reveal opportunities and challenges for children's inclusion and learning approaches, informing recommendations for promoting inclusive play in ECEC. Inclusive play emphasizes diverse strategies accommodating differences in learning styles and modes of knowledge expression among teaching staff and children.

**Keywords:** inclusion; play; case study; early childhood education and care; pedagogy

## 1. Introduction

The emphasis on play is one of the acknowledged strengths of Finnish inclusive early childhood education and care (ECEC) [1]. In Nordic countries, it has an important role in the promotion of children's holistic learning and well-being during the early years before primary education [2–5]. Inclusive values are one key element of ECEC policies in Finland. Inclusive education is understood as being equal and based on participation opportunities and individual support for all children [6–8].

It has been shown that pedagogical play is the most important method for supporting the participation and learning of culturally and linguistically diverse children [9,10]. Play itself is an activity where children imagine and practice, and where children work on designing and focusing goals at an appropriate level of challenge [11,12]. Based on the inclusive principles described in ECEC guiding policy documents, teaching staff should provide opportunities for diverse play to all children, guide children's play in versatile ways, and scaffold communication and language development through play. Furthermore, the staff should ensure that each child can experience participation in shared play equally, according to their competencies and skills [5,6]. However, previous studies about the practical implementation and practices of playful activities and pedagogy, or inclusive play activities in ECEC, are scarce.

Therefore, in this study, the focus is on the practical elements and methods teaching staff are implementing and children are experiencing during playful activities and play in ECEC classes in Finland. We frame play as a pedagogical activity scaffolded by teaching staff using the model of playful pedagogy by Kangas and Harju-Luukkainen [2]. We aim to focus on children's experiences in inclusive ECEC settings through play. We explore inclusive play pedagogy in the context of culturally and linguistically diverse ECEC. We aim to build understanding and give effective support examples about how shared play between staff and children enables meaning making, creates mutual cultural knowledge, and thus supports children's belonging to the community.

Our data are part of a long-term study, "Keys for Shared Understanding", which examines the implementation, best practices, and further needs of the development of inclusion, diversity, participation, and support for children and families of culturally and linguistically diverse backgrounds in the Finnish ECEC. The project focuses particularly on inclusive and participatory pedagogy, play, interaction, and a language-, culture-, and worldview-aware learning environment.

The data for this study were gathered in two ECEC centers with culturally and linguistically diverse teaching staff and children. The research data were collected through everyday video observations, non-participating observations by researchers, and research diaries. The research objective is to explore the elements of playful inclusive pedagogy and further answer the research questions:

1.  What kind of playful practices create opportunities for practical inclusion for children?
2.  What kind of shared meaning-making processes emerge between children and teacher(s) in these playful practices?

Next, we will introduce the theoretical background of this study to clarify the concepts of inclusion and playful pedagogy in the Finnish ECEC context. In the methods, we explain in detail the data collection and analysis. In addition, ethical questions are critically discussed to highlight the importance of the well-being of participants, particularly of young children. The findings are reported with a discussion on the identified elements of inclusive play and recommendations for practice. In the end, the conclusions are presented to outline the broader possibilities inclusive play can offer.

## 2. The Finnish ECEC Highly Promotes Inclusion

Inclusion in education has a variety of definitions and frameworks depending on how and on which orientation it is considered. In Finland, it has been highlighted [13] that inclusive education is not mentioned nor defined in education legislation, therefore creating inclusive myths and variations in practices. The concept of inclusive education is characterized by its evolving and transformative nature, offering unobstructed entry to all individuals, and embracing a perspective that perceives possibilities rather than limitations [7,9,14].

The framing of inclusion through human values involves considering education policies through the Salamanca Statement [15] or through the declaration of human rights, where all people should be treated equally and with respect [16]. These policies are followed in all Nordic countries, including Finland [3,6]. Alternatively, examining inclusion through the lenses of learning theories entails tailoring education to the individual and prioritizing personalized learning for those requiring additional support and attention [17]. This approach underscores the importance of catering to diverse learning needs and effectively embracing an inclusive environment that is tailored to suit the diverse learning requirements of each child. However, these two approaches may seem dualistic, offering inclusion for all in policies but then offering solutions only for individual support. We focus on inclusion as a more varied set of values, tools, and methods in community-level practices in ECEC, where all the children in a group or a center could benefit.

In the Finnish ECEC, inclusion is seen as a principle, a value, and a holistic way of learning. The inclusive principles are equality, equity, non-discrimination, appreciation of diversity, social participation, and togetherness. ECEC staff should implement these

values in inclusive pedagogy in which each child can participate together with other children regardless of the need for support, disability, or cultural background [5,6]. These diversities make each child and family unique, as everyone is seen as possessing special resources for the ECEC community [6,9,11]. The essence of inclusion lies in recognizing, hearing, and considering the voices of all individuals. This reimagining of inclusion as a foundation for Finnish ECEC hinges on values such as equality, equity, and the inherent rights of children [3–5]. However, in recent years, the critical question of inclusive practices and pedagogical tools, including tools for hearing, understanding, and creating shared meaning making, as well as enhancing belonging and inclusive spaces in the everyday practices of ECEC, has been raised [17–19]. According to the research on teaching skills [17], the competencies of special needs, diversity, special education, and multi-professional cooperation of early childhood teaching staff need to be strengthened.

*Contextualizing Playful Pedagogy in the Context of Inclusion*

To understand what play entails for individual children, it is essential to define and perceive play as a social activity contributing to children's social development. According to Claughton [20], children are active social agents, and their engagement in child-led play leads to intrinsic learning opportunities. In a general sense, play, when scaffolded through active support and an appropriate learning environment in ECEC, can be seen as a societal path for a person in society [21].

The Nordic pedagogical tradition places a strong emphasis on play through a focus on holistic development and continuous learning through varied objectives [2,4,21]. Play and academic learning are not seen as opposites, but through play, the learning of cognitive topics and skills is also possible [22,23]. This aligns well with the values of inclusion, where the participation and involvement of each child have been seen as key elements, for example, in the Finnish National Core Curriculum for ECEC [5] (p. 9) which states as follows: 'The purpose... is to create equal preconditions for the holistic growth, development, and learning of the children...'.

However, the research [7] shows that in the practices of ECEC, culturally and linguistically diverse children have fewer non-social roles, fewer participative actions, and less competence to participate in role- and imaginary play. Still, it has been identified [10] that the support of play is the most important way to increase the participation and learning of culturally and linguistically diverse children [9]. Furthermore, dialogue, support, care, a sense of belonging, mutual understanding, play pedagogy, the promotion of knowledge, competence, and strengths, equal interaction, active participation, the importance of other children and families, and positive emotions have been shown as crucial elements when building an inclusive and participatory ECEC pedagogy [6,8,24].

Playful pedagogy is a concept defined by Kangas and Harju-Luukkainen [2]. It combines learning theories with the conceptualization of play as children's motive, natural activity, and focus and provides a framework that underscores the gradual development of play and learning through pedagogical scaffolding; see also [21,25,26]. Pedagogical play involves teachers' observation, interaction, support, and guidance [2]. It aligns educational activities with curriculum goals through playful interactions and nurtures creative exploration, independent initiative, and goal setting [21,27]. Through the playful pedagogy model, it is possible to support individual children's development in communication, interaction, cognitive abilities, self-regulation, and competencies related to children's joint meaning making, problem solving, and creativity [2].

Embracing individual interests and shared representation in play and scaffolding the holistic activity can be done by moving away from the concept of cosmetic play and notions that certain types of play are inappropriate [28]. Play is defined by a child's interaction and intrinsic motivation, and engagement with peers or adults enhances children's knowledge and abilities [29] as children explore their environment and engage in playful interactions to build their knowledge and experiences [20].

By combining play-focused research with studies of special education and childhood studies, a more comprehensive conceptualization of children and their play emerges. This perspective portrays children as cognizant, intentional, and capable participants in play [11,21,28]. Recognizing that children actively construct knowledge about their world, it becomes vital to provide support that builds upon children's self-awareness, implemented politically and with an emphasis on play as well as on the child's activity and participation [5,30]. Previous studies have shown the effectiveness of concrete activities in promoting participation and inclusion, particularly among culturally and linguistically diverse children [10]. Activity emerges as a catalyst for continuous involvement, with a specific focus on sustaining intense engagement and effectively boosting participation, creating stronger social interactions, and contributing to inclusion, particularly for culturally and linguistically diverse children [8,14,21].

Inclusive education emphasizes the implementation of diverse strategies that embrace differences in learning and modes of knowledge expression. Educational institutions continuously explore inclusive education approaches to teaching with a focus on universal design and differentiation to ensure that the classroom environment and curriculum are accessible to all [20]. Therefore, there is a need for the development of an understanding of inclusive play and contextualizing the supporting pedagogy to promote inclusive education in which all children and staff members have resources and tools to create together shared meanings, mutual understanding, equal learning opportunities, and experiences [2].

## 3. Methods

The research is a case study in nature and was conducted in the spring of 2023 in ECEC centers. In this study, we use data from two of these centers. The research data were collected through everyday video observations, non-participating observations by researchers, and research diaries. The case study approach is suitable when a phenomenon in focus is concrete, and the study seeks to obtain a contextual and in-depth understanding of a specific subject in real practices or environments [31,32]. The case study was particularly used in this research to carefully examine the complex phenomena of inclusive education practices [32].

The primary objective of this study is to investigate the components of a pedagogical approach characterized by playfulness and inclusivity. This exploration is driven by the pursuit of answers to the following research questions: (1) what kind of playful practices create opportunities for practical inclusion for children? And (2) what kind of shared meaning-making processes emerge between children and teacher(s) in these playful practices?

To discern the manifestations of these playful inclusive pedagogical elements in authentic ECEC settings, a multifaceted research approach was employed. This approach encompassed everyday video observations, non-participatory observations conducted by the research team, and the maintenance of research diaries. The utilization of video observations served as an essential methodological strategy for capturing the naturally occurring instances of playful practices in real-world ECEC scenarios. These recorded interactions enabled a nuanced exploration of micro-moments, which could not be authentically conveyed through alternative research methods.

Furthermore, it is imperative to highlight that the triangulation of methods was instrumental in enhancing the reliability of the study's findings. Specifically, the convergence of data collected through video observations and the insights documented in research diaries, compiled by a team of three researchers, contributed to the robustness of the research outcomes.

### 3.1. Data Collection

The study was conducted through video observation and recording of children's play and interaction situations with teaching staff. The ethical questions of conducting data in the everyday learning environment of children and teaching staff were paid careful

attention [33]. There were five ECEC groups with 10 teachers and 21 children in each, and the children were between 2 and 7 years of age. There were five mornings filmed, each of those in different groups. Filming and research permits were requested from all the children in the group, but only 4–6 children were monitored at the same time. An important aim of the study was to develop a methodology that makes it possible to study the inclusive practices of play transparently and to address children's experiences, considering them to be pedagogical practices [34]. The video cameras were movable, and researchers controlled them with their wristwatches. The teachers were free to organize the class, so in the groups the children were focusing on self-initiated play. The researchers used the camera to follow the play. Videos were reviewed on the day of filming, files were named informatively, and keywords, dates, and times were recorded in the observation matrix. The total amount of research data covered twenty hours of video data, approximately three hundred minutes from these two ECEC centers. For this study three play cases were selected for closer narrative analysis (Table 1) to highlight the inclusive play practices and children's experiences in these.

**Table 1.** Research data: participants and durations of the video clips.

| Video Clips | Case | Participants | Duration | Total Length |
|---|---|---|---|---|
| clips 1–2 | Pizza Chefs | 3 children (5–7 years old, boys) 1 teacher | 05 min 37 s 00 min 47 s | 6 min 23 s |
| clips 3–9 | Awaken | 2 children (2–3 years old, girls) 1 teacher | 02 min 20 s 00 min 17 s 03 min 25 s 01 min 13 s 04 min 36 s 06 min 19 s | 17 min 20 s |
| clips 10–11 | Postman | 5 children (5–7 years old, girls) 1 teacher | 04 min 32 s 01 min 57 s 01 min 29 s | 7 min 18 s |
| 11 | 3 | 10 children, 3 teachers | | 32 min 19 s |

*3.2. Data Analysis*

Methodologically, the foundation of the analysis rests on a narrative analysis approach. The realm of play is inherently rich in stories, layered fantasies, and imaginative constructs intertwined with tangible actions, communication, and interactions among participants [29]. This rationale guides the selection of the narrative approach to underpin and elucidate the children's experiences and activities within the collected data. Our endeavor during the analysis process was to comprehensively construe and articulate the children's experiences in a manner that respects their meanings, intentions, and actions as manifested within the context of inclusive play [7,8,21]. Data were analyzed through content analysis using the researchers' triangulation to discuss the visual communication and embodiment themes. Materials were transcribed, and the analysis was stored in the theme matrix. In this pursuit, we employed the abductive approach, a method characterized by systematic creative reasoning in research aimed at generating novel insights [35] all the while valuing and acknowledging the authentic voices of the participants. The abductive approach is based on the interaction between data and theory through which it aims to provide novel discourse and new theory concerning the research topic [36]. The names of participants were encoded for the analysis process.

The video data for this specific study's narrative comprise 11 short video clips, totaling 32 min and 19 s (including all the data of these cases). These video clips were subjected to a coding process, wherein they were assigned code names such as 'expressing an idea', 'spoken interaction',' kinesthetic communication', or 'expressing an idea of action'. The coding was done twice, first focusing on the interaction between teacher and children and then more carefully coding the verbal and non-verbal initiatives and expressions

of the children interacting with peers, tools, or the environment. The coding procedure was implemented in conjunction with relevant theories, employing abduction to generate systematic output phenomena [36].

From the pool of encoded and categorized findings, a careful selection was made highlighting three narrative excerpts that effectively encapsulated the primary discoveries of the study. The exploration of children's interactions and communications spanning various activities took the form of a narrative inquiry. This inquiry delved into the unique ways in which children manifested their expressions, actions, communication, and processes of sense making.

*3.3. Ethical Questions*

An observational study always consists of some ethical issues. The researchers aim to understand and see the daily experiences of children who need to involve themselves in the activities of the ECEC center. The starting point is the interest of the participants in ECEC. For example, the study by Frankenberg [37] examined the ethics of observing children in a very comprehensive way. It informed all participants that the role of the teaching staff, children, and guardians who participated in the study was to help the researchers conduct research instead of being objects of the research. Also, in our research, the aim was to support teaching staff, parents, and children to feel safe and secure, participate in a joint, shared effort of data collection, and have an active role in supporting all participants' experiences of inclusion and agency.

There has been criticism [38] towards research on diversity when researchers choose particular groups and thus regard those groups as different and exceptional, or take predefined categorizations of children to reproduce dominant norms. Therefore, it is pivotal to stress that in this study, by choosing to focus on culturally and linguistically diverse children's play, we have not excluded other children. Moreover, we see that inclusion encompasses all children, families, and teaching staff. The research was prepared and implemented with good scientific research practices [33]. In this study, these ethical guidelines meant that the researchers respected the dignity, autonomy, and rights of children: the children participating in the study were able to participate voluntarily, but also to refuse to participate or discontinue their participation at any time without experiencing any harm or negative consequences. The children's rights, such as self-expression, feelings, or social interaction, were not affected in any way during the data collection of the study. The children's privacy was respected by affecting their day in the ECEC center as little as possible [39].

The ethical review statement was admitted by the Research Ethics Advisory Board of the University of Turku. In addition, the special features of the participation, experiences, and opinions of the children and guardians were also considered in the research.

## 4. Findings and Discussion

Firstly, we introduce the selected play events that happened in the spring term in three different child groups. We describe the events as narratives to give a holistic vision of the play. Each child group had a majority of children with a non-Finnish-speaking background, and their ages varied between 1.8 and 7 years. All the selected events focused on child-initiated play with more than one participant and a teacher participating in play. Participants' actions, expressions, and interactions are described carefully together with the spoken communications to allow the readers to investigate the actual event itself. The existing language is preserved as well as possible. For example, grammar mistakes or bilingual expressions are explained in the text with language name codes [Finnish] and [English]. If the language was not recognized, code [other] was used. The names of the children are pseudonyms; we used the letters *a* to *i* to give code names to each participant.

Secondly, we explain and describe the analysis categories of practical inclusion: teachers' participation in play, flexible design and adaptive goals, playful language, enabling tools with non-verbal and kinesthetic communication, and emerging play with interaction

and long-lasting intensity. From these, we identify both the opportunities and challenges for children's inclusion through participation and learning approaches and suggest recommendations for supporting inclusive play in ECEC.

*Event narrative: The Pizza Chefs*

Three boys with non-Finnish-speaking backgrounds played a cooking game around a table with kinesthetic sand and baking tools. The teacher had set a pictorial list of items with Finnish language vocabulary on top of the table and actively used it to repeat the vocabulary concerning cooking tools and the names of the imaginary food.

The children interacted with each other using short sentences but also mimicked and used kinesthetic communication. One of the boys, Amez, took a sand-pie in his hand and said, "Wait! I will eat this." [Finnish], mimicking eating hot and delicious food with his facial expressions and gestures to Benjamin, and the children laughed. Amez gestured that the pizza was hot, opening his mouth wide and exclaiming, "Hot! Hot!" [Finnish]. Benjamin half-stood in his chair and laughed at Amez. Josef grabbed the plastic bucket with both hands and pretended to drink from it. Benjamin focused on baking his pizza, and the teacher asked Benjamin, "Did I get that salami pizza yet?" [Finnish]. Benjamin concentrated on baking the pizza and quietly replied, "Wait. Not yet." [Finnish]. The teacher confirmed to Benjamin, "Wonderful! Mmmm—" and instructed the third boy, Calah, to clean his play tools in the box.

Calah was ready to leave, and the teacher went with him to the lobby while Benjamin and Amez continued baking and building a huge pizza in the sandbox on the table. Benjamin grabbed the toy knife and started cutting the pile of pizza sand enthusiastically, whereupon Amez joined in, holding the box and occasionally holding Benjamin's wrist, whining, "Oh—pizza slices" [English]. Benjamin cut with concentration and said partly to himself, "Thank you" [English]. Amez baked a smaller pizza and commented, "A small pizza" [Finnish]. Then he rose and started dancing. Amez asked the dancing Benjamin for cheese on his pizza, "Cheese! Look at me, look at me—it is for you!" [English] and put something (sand) on Benjamin's pizza. Benjamin reacted by nodding and stating, "Jeez" [Finnish or English]. Amez got up from his chair to cut the large pizza, saying, "Okay!" [English].

The teacher popped back into the room and said, "Hey pizza cooks, you have five minutes for pizza chefs to make pizza." Benjamin grasped the term pizza chef and pointed first to Amez and then to himself: "We are pizza chefs!" [Finnish]. The teacher happily confirmed, "Well, you are pizza chefs!" [Finnish]. Benjamin tried out the concept, "Because we did it—We did it". Amez spoke over him, "We were going to—we were going to do this cuzi" [Finnish] [other]. Benjamin and Amez took a new concept and spoke in unison: "Yeah. Let's do this cook[ing]. Let's make cookies." [Finnish, then English]. Benjamin confirmed by nodding "Okay" and turning back to Amez, "No, we—we are the cooks" [Finnish]. Josef nodded while mixing the pizza dough with his whole body and supported Benjamin, "Yeah, cook, cooks!" [Finnish]. Benjamin elaborated: "Cooks are not like that." Benjamin took some sand on the table and shaped it into a small pile. "Look. This is how cooks should be" [Finnish] and showed how the chefs make pizza. Amez closely followed Benjamin, who patted the dough against the table and announced, "That's it. I saw" [Finnish] and then replied, "Yeah. This is what I do" [Finnish].

Five minutes later, the teacher ended the play, and the boys started cleaning up.

*Event narrative: Awaken*

Children in the toddlers' group were assembled in a playroom. Two-year-old girls Danilla and Esha, along with a teacher, were engrossed in play involving dishes and dolls. The teacher sat on the floor next to Esha, and followed the play.

Danilla briskly approached the teacher, offering a plate through kinesthetic communication, and said, "This is yours" [Finnish]. The teacher echoed, "Is this for me? Thank you." Danilla added, "For my mother. For my mother," with the teacher responding positively, "For your mother?" Esha, upon hearing the word "mother", chimed in, confirming, "My mother is coming" [Finnish]. Esha's statement garnered a sympathetic repetition from the

teacher: "Your mother is coming too. Good thing." The teacher then engaged in eating the offered play food through kinesthetic communication, diverting the children's attention from the topic of mothers.

Danilla, leaving a baby doll on the floor, turned to the teacher to communicate through kinesthetic gestures, saying, "Baby, eat—my baby eats" [Finnish] [English]. Esha joined Danilla and vigorously stirred a coffee cup with a spoon while observing Danilla's actions with the baby doll. The teacher acknowledged Danilla's actions with nods and facial expressions, inquiring, "Is the baby eating?" Danilla and the teacher repeated the words "eat" and "baby," with Danilla returning to the baby doll to examine its face, declaring, "No, baby eats" [Finnish]. Although the teacher did not acknowledge the incoherent grammar, she asked Danilla about the baby doll's diaper while pointing to it, saying, "Does he have a diaper?" [Finnish]. Danilla, however, did not respond to the question. Instead, she attempted to open the baby's eyes and enthusiastically declared, "Awake! Awake!!" [Finnish]. The teacher concurred, saying, "He's awake," and Danilla picked up the baby doll, repeating, "Awaken. Awaken. Look! Aah, awake!" while expressing joy and wonder through her facial expressions and gestures. The teacher affirmed, "Oh. It is awake," and both shared laughter. Esha quietly moved away and began selecting cutlery.

While Danilla continued to laugh, the teacher introduced a new element to the play, asking, "Do you want to give him food—or milk or anything?" [Finnish]. Esha returned to Danilla, reiterating "milk" [Finnish], and crouched down beside the baby doll. Danilla also repeated "milk" [Finnish] and cradled the doll in her arms. Esha spun around on the floor with a cup of coffee in her hand and presented it to the teacher, inquiring, "Wanting milk? Wanting milk?" [Finnish]. The teacher graciously responded with "Thank you!" [Finnish]. When Esha's spoon fell, Danilla, still holding the baby doll, bent down to retrieve it and carefully placed it in the cup held by Esha. The children did not engage in direct verbal communication.

*Event narrative: Postman*

In a group room bustling with many children, Fiona and Gerli, two girls, had constructed a narrow hut out of mattresses and observed the activities in the classroom from behind it. The teacher sat next to the hut and drew the children's attention to it: "Shall I give you some mail here?" [Finnish] she asked, pointing with her finger to the gap between the two mattresses [kinesthetic communication]. The question caught the attention of the girls, and they both turned to the teacher. Fiona replied, "Yeah—our mail." [Finnish]. Gerli dove under the mattress to look inside the mailbox, and Fiona continued the conversation with the teacher. The hut was so small that only one could fit next to the mailbox at a time. The teacher asked, "Did you order any post parcels, or do you want envelopes—letter mail?" [Finnish]. "Letter mail," Fiona confirmed, but the teacher did not hear and continued, "Or did you order princess dresses?" [Finnish]. Fiona jumped up, looking excited [expressions and gestures], and confirmed, "Mail and dresses!" [Finnish] The teacher replied surprisedly, "Oh. Letter mail and dresses. Okay, I'm going to the store now to buy the supplies." [Finnish]. The teacher got up and went to the other side of the room. Gerli emerged from under the mattress, but the teacher was already gone, and both children waved [kinesthetic communication]: "Bye-bye!" [Finnish].

The children were left hanging inside the hut, and the play did not really start. Fiona chatted to herself and climbed on the pillows restlessly, and Gerli showed the researcher the toy she found on the floor. After a while, the teacher returned, and the children's attention was directed back to her. The teacher said, "Now there would be mail! Here comes a letter for the gentry." [Finnish], and she put the paper in the hole between the mattresses. Fiona got there first and crawled under the mattress to receive the letter, exclaiming, "Thank you!". The teacher continued: "You had also ordered a princess dress." [Finnish] and started threading the dress between the mattresses while continuing to say, "This is a parcel delivery without postage." [Finnish]. "Then you will get yellow sunglasses," the teacher continued to tell Fiona slipping the next item into the mailbox, where Fiona continued to receive the delivery. The teacher added, "And the yellow crab you ordered." [Finnish].

Gerli spun round to the other end of the hut and did not seem to react to the continuation of the play. Finally, she also became excited about the delivery and jumped up and down and tried to show the crab she had, but the teacher's attention was already directed to another child, Habibah, and she started talking to her. All in all, the delivery took 25 seconds and was over before Gerli had time to join the game.

Soon Fiona and Ibtisam were excited to play with the newly delivered toys, and the mail delivery game was between Fiona and the teacher, whom Habibah followed curiously. Then Habibah joined the game and negotiated with Fiona about the next mail delivery. Simultaneously, Gerli, who finally had room to address the postman, ordered crayons from the teacher by showing the letter in her hand [kinesthetic communication] and saying, "Pen...yellow." [Finnish]. The teacher assured her, "Would you like to order pens? A yellow pen?" [Finnish] and received a nod and a smile [expressions and gestures]. The teacher left to pick up the order, and Fiona and Habibah got up to notice that the postman had already moved on. Together, they continued their conversation with their backs to the camera.

The teacher came back soon and stated, "This is the last postman delivery." [Finnish], knocked [kinesthetic communication] and said, "Here the postman brings you a box of markers." [Finnish]. Gerli accepted the markers and sneaked back to the other end of the hut. Habibah, in turn, sneaked to the mailbox. However, the teacher did not continue the Postman interaction with the child because another adult's voice drew her attention.

### *4.1. Elements of Inclusive Play*

Based on our analyses that are presented through these three selected excerpts, the following elements of inclusive play were identified. These elements are: (1) teachers' participation and active presence, (2) repetition with a flexible plan and adaptative goals, (3) playful language and joy of play, (4) enabling tools with non-verbal and kinesthetic communication, and (5) emerging play with interaction and long-lasting intensity. Next, these elements of inclusive play will be discussed with recommendations and suggestions for inclusive practices.

#### 4.1.1. Element 1: Teachers' Participation and Active Presence

The teachers' direct support emerges through face-to-face communication and the expression of gestures and facial expressions while playing with children or observing them closely. The teachers were trying to support the play by talking play language, taking a role character, being actively present for play, and even leading the play. However, while actively supporting the play, the teachers in many of our recorded video data episodes seemed to miss children's nonverbal and even verbal communication, as in the Postman excerpt. On the other hand, in the Awaken excerpt, it is visible that the teacher, who has positioned herself on the floor in the middle of the play area, could observe and react to different styles of communication. It seems that the need for support varies during play, and while teaching staff are focusing on one element of support, they do not pay attention to other means of support.

Based on the result, it is suggested that awareness of different means of communication—verbal, expressions, gestures, and kinesthetic communication—should be strengthened to provide teaching staff with the tools to respond [10]. Participating in the play for teaching staff is also essential; therefore, it is important that teaching staff share responsibilities in diverse ECEC groups so that at least one teacher can fully focus on participating in play with the children. At its best, the active role of teaching staff in play creates spaces for sharing meaning making in play that children express through joy and laughter.

#### 4.1.2. Element 2: Enough Repetition but a Flexible Plan and Adaptive Goals

In the Awaken excerpt, as in some other data clips, it was visible how the teacher was trying to guide the children to follow a plan or routine in play. In the Awaken excerpt, the teacher supports the children in taking care of babies following certain events, such as feeding, changing diapers, etc. However, the children are not interested in this, but

the teacher tries to repeat the same advice to them repeatedly, which prevents her from registering and responding to the children's other initiatives and ideas.

While children with diverse backgrounds and varying ages are playing together, the plan for play cannot have too many set goals in order to give room for children's imagination, initiative, and participation. Therefore, we see that, without set goals, there is room for shared meaning making to emerge through exploration [23]. Some children could need opportunities for the repetition of concepts, ideas, and plots of play several times, and play design should be clear enough to not exclude anyone or prevent participation. Our results suggest that if the teaching staff have opportunities to organize playgroups, they could, from time to time, choose particularly those children needing more time and support for conceptualization and practice with language in play through interaction and meaning making. On the other hand, it is also recommended to encourage children to play with more competent peers to enable learning in the zone of proximal development; see also [12].

### 4.1.3. Element 3: Playful Language and the Joy of Play

Playful language is an element both teachers and children use during play activities. It gives room for funny errors and playing with concepts without increasing the stress of failure [27]. In the Pizza Chefs excerpt, the children get a new concept, a cook, from the teacher, who does not seem to first pay attention to the concept itself, but when the children begin to explore and contextualize the concept, the teacher gives positive and warm feedback for them. Soon, the children develop ownership of the new concept and joyfully express, "We are the pizza chefs!". A similar conceptualization takes place in the second episode, where the concept of 'awake' arises in joy and involvement. The playful language gives opportunities to practice the concepts in versatile ways and for the repetition of more alien concepts to build ownership.

Based on the results of this study, it is recommended that teaching staff should use playful language and humor in play activities [40] and react when children find something hilarious by joining in the laughter and joy. If the use of playful language is challenging, teaching staff could use puppets and play tools to increase the amount of it. Language is not only vocabulary and grammar, but different means of communication are elements in the path of socialization [21].

With culturally and linguistically diverse children, it would also be recommended to use children's diverse linguistic repertoires as resources in play; see also [41]. One example is the approach of "funds of knowledge" [42] in which children's and their families' diverse languages, traditions, and interests are considered as resources for educational practices. In play, children can integrate the cultural experiences of their families but also those from the ECEC community allowing them to engage meaningfully [43,44]. This requires the teaching staff to have the competence to employ play as a method for incorporating children's diverse knowledge and use it for shared understanding and a sense of belonging [45].

### 4.1.4. Element 4: Enabling Tools with Non-Verbal and Kinesthetic Communication

Children's communication and actions of play are kinesthetic and visible in the main findings of the data. Young children tend to show this with their hands and embodied communication, where the teaching staff need to observe the movement, use of tools, gestures, expressions, and emotions of children, as well as focus on the spoken words. This requires a high-quality understanding of children's development and observation skills from the teaching staff [22]. In the Postman excerpt, the teacher, sitting behind the mattress, is not able to see kinesthetic messages from children or does not always see who is behind the mailbox. Thus, some elements of communication remain unnoticed, and some children are excluded from the play. Also, in the Awaken excerpt, the teacher focuses on the more active child, while the other child behind her back does not get support for her non-verbal initiatives. On the other hand, the teacher reacts and confirms the initiative the children express by using tools as a means of communication.

Based on the results of this study, kinesthetic and non-verbal elements of communication should be part of the competence of teaching staff. The tools used with children of a young age or from diverse language backgrounds should be realistic and actively used to support the development of vocabulary. For play design in the classroom, it would be important to have enough room to avoid situations such as the Postman excerpt where all children did not have room to participate in the play; see also [14].

4.1.5. Element 5: Emerging Play, Interaction, and Long-Lasting Intensity

The event taking place in the Pizza Chefs excerpt, where the play starts to emerge simultaneously when the teacher is already planning to end the playtime, was common in many other video data episodes too. While the children do not have a shared language to plan their play beforehand, the play needs time and space to gain its full potential and continue to have a long-lasting intensity [21]. Even in planned play sessions, the play is emerging, and thus teaching staff cannot know the results or flow of the play.

We recommend, based on our study, that it is important to reserve enough time and observe the children's peer interaction and level of excitement before interrupting the play. Also, in the Postman excerpt, the teacher ends the interaction by announcing the final delivery, while some participants have not had time to participate in play or it has taken time to agree and negotiate between the children about the flow of play. While children represent diverse cultural and language backgrounds or different ages, peer interaction takes time [17]. Our results suggest that teaching staff should support the play pedagogically through long-term interaction and slower interaction events taking place in the play. It is recommended that more time for emerging play and children's peer interaction is provided.

To sum up the elements of inclusive play, we suggest several recommendations for scaffolding children's interaction, learning, and meaning-making processes by teaching staff:

- Teaching staff's awareness and competence of different means of communication—verbal, expressions, gestures, and kinesthetic communication—in play should be strengthened.
- Teaching staff should actively participate in playful and play activities with children.
- Smaller playgroups, particularly for those children needing more time and support for conceptualization and practice with language, should be used.
- Children should be encouraged to play with more competent peers to enable learning in the zone of proximal development.
- Teaching staff should use playful language and humor in play activities, and react when children find something hilarious by joining in their laughter and joy.
- Teaching staff could use play materials and tools to provide opportunities for playful language use.
- Play could be used as a method for incorporating children's diversity for creating shared understanding and a sense of belonging.
- The tools used with children of a young age or from diverse language backgrounds should be realistic and actively used to support the development of vocabulary.
- Teaching staff should reserve enough time and observe the children's peer interaction and level of excitement before interrupting the play.
- Teaching staff should support the play pedagogically through long-term interaction.

*4.2. Discussion*

In education policy discourse, it has been agreed in the Nordic context that inclusion is a key value of ECEC [6,8,10,41]. However, the methods and tools for inclusive practices and scaffolding for children's everyday inclusion have remained scarce and have been focused on individual special needs support. Nevertheless, for children with diverse cultural and language backgrounds, play is predicted to be a tool for promoting inclusion and inclusive pedagogy [8,11]. Our findings suggest that elements of inclusive play could

be implemented to provide opportunities for practical inclusion and to involve members of the ECEC community in shared meaning-making processes.

As researchers, we see that play itself entails diversity: it might be physical or disembodied, full of different emotions, humoristic or serious, loud or quiet, verbal or non-verbal, creative, musical, imaginary, voluntary, joyful, non-intentional, absurd, empowering, healing, holistic, and collective or individual; see also [11,29]. We regard play with its innate ability to transcend, e.g., cultural, linguistic, and physical barriers in ECEC, as a unifying force that brings diverse children and teaching staff together if inclusive play pedagogy is applied. For example, play is found to be a tool for teaching staff to participate in child-initiated activities and allocate support for children's needs through playful language, kinesthetic communication, or right-timed interaction.

Play also offers experiences and links the curricular elements to children's everyday events in ECEC and wider society and to scaffold children's societal path, well-being, and sense of belonging in a community [18,21,22]. The approach of playful pedagogy views the play as a multimodal experience and as an attitude towards the world, and it gives teaching staff tools to increase the value of play. We have suggested some recommendations based on the observation data and children's experiences in diverse group-based activities in Finnish ECEC. The recommendations could serve as tools for aiming the support and scaffolding in play to focus on the challenges of inclusion. In the context of ECEC, play's intrinsic value and its pedagogical significance are recognized, while the ideology of the playing–learning child places play at the core of education [2,20,25]. This study's contribution is that through choosing, implementing, and enabling different elements of inclusive play based on observations and children's needs, teaching staff can aim to build a culture of participation and well-being in ECEC.

## 5. Conclusions

This study offers valuable results on inclusive pedagogical practices through play, indicating that inclusion is connected to participation in an educational context. Earlier Nordic research results [8,41,46,47] underline that ECEC staff can understand linguistic and cultural diversity by sharing play-based activities with the children with a focus on participation, belongingness, and engagement. However, models of supporting learning and participation at a practical level have been scarce. The understanding of playful learning and teaching illustrates how play is understood as a tool to promote inclusion in different ECEC contexts [2,9,11,48]. In the future, it would be important to research pedagogical opportunities for inclusion in ECEC and further contextualize the use of play and playful learning to support children with equal opportunities to express themselves and join in shared meaning making.

Inclusion and a sense of belonging are deeply intertwined, forming the bedrock of participatory pedagogy through the many languages in which children express themselves. Play serves as a profound method and tool, fostering children's learning, well-being, active participation, and collaborative meaning making within the realm of ECEC [2,4,21,29]. Inclusive education seeks to recognize diversity as an asset, fostering a socially just environment. Incorporating inclusion into the heart of Finnish ECEC represents a profound shift that aligns with participatory pedagogy, equity, and respect for diverse voices [3,7,9,48]. As a key element of ECEC, we propose that play could be seen as a pathway between the individual and broader political approaches to inclusion.

**Author Contributions:** Conceptualization, J.K., A.-L.L. and O.A.; methodology, O.A. and J.K.; software, J.K., A.-L.L. and O.A.; validation, J.K., A.-L.L. and O.A.; formal analysis, J.K.; investigation, J.K., A.-L.L. and O.A.; resources, J.K., A.-L.L. and O.A.; data curation, J.K., A.-L.L. and O.A.; writing—original draft preparation, J.K., A.-L.L. and O.A.; writing—review and editing, J.K., A.-L.L. and O.A.; visualization, O.A.; supervision, J.K.; project administration, J.K., A.-L.L. and O.A.; funding acquisition, J.K., A.-L.L. and O.A. All authors have read and agreed to the published version of the manuscript.

**Funding:** This research was funded by the Jenny and Antti Wihuri Foundation, Open access funding provided by University of Helsinki.

**Institutional Review Board Statement:** The study was conducted in accordance with the Declaration of Research Ethics Committee Finland (TENK), and approved by the Institutional Review Board (or Ethics Committee) of the University of Turku (6.2.2023).

**Informed Consent Statement:** Informed consent was obtained from all subjects involved in the study.

**Data Availability Statement:** Data is unavailable due to privacy reasons.

**Conflicts of Interest:** The authors declare no conflict of interest.

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
