# Peer review of "Inclusive Play: Defining Elements of Playful Teaching and Learning in Culturally and Linguistically Diverse ECEC"

_education, doi:10.3390/educsci13090956_

Round 1

Reviewer 1 Report

Thank you to the authors for their research in this area and their work on this paper. Play-based pedagogies deserve continued & increased research, so it was lovely to see this paper come through for review and I was glad to have a chance to read through it.

I hope the following feedback is helpful in further developing the paper:

* Line 32: can you please further explain "ideal level" as this is somewhat vague

* Lines 56-60 are somewhat repetitive, the writing could be refined here so that it is sharper and clearer.

* Lines 92-94 -- the sentence re: equity & children's rights should be substantiated w/ a citation or two.

* The ethics statement is good and it is lovely to see this addressed in detail. I would recommend some further detail around a few points e.g. the part about "good scientific practices" - can you capture these in brief? Can some further citations be woven into lines 218-227? There is some terrific literature on ethics in ECEC research and ethical issues when working with children & families, this would really strengthen this section.

* re: the children's ages, this is inconsistently reported (see line 175 vs line 242). This also should be reported in  more detail. Could participant profiles be provided, with pseudonym, age, some info about each child, in a table format? This would be very helpful for the reader to learn about the cohort and the context.

* findings, discussion, and conclusion are well written, comprehensive, and insightful. Well done to the authors on clearly illuminating the children's play and the significance of these encounters. They have also done well in making clear the contribution of their study. 

* Copy-editing needed throughout, esp. in the abstract.

Some editing is needed, in particular there are some tense issues throughout and some typos in the abstract.

Author Response

Thank you for the revision for the encouraging feedback and important insights on our manuscript! As researchers, we were happy to read about the strengths of our study - a sincere thank you!  Here are the detailed answers for the suggested corrections and revisions. We hope we have succeeded in developing the paper. 

Reviewer 2 Report

TITLE AND ABSTRACT

Title Evaluation:

Clarity and Specificity: The title, "Inclusive Play: A Finnish Case Study in a culturally and linguistically diverse ECEC setting," effectively conveys the topic and context of the study. However, it could be made more specific by briefly mentioning the key findings or outcomes. This would give readers a clearer idea of what to expect from the article.

Abstract Evaluation:

Clarity and Conciseness: The abstract provides a clear overview of the study's purpose, methods, and findings. However, some sentences are quite long and could be broken down for better readability. For instance, "Inclusive values are one key element of early childhood education and care (ECEC) policies, and inclusive education is understood as equal and based on participation opportunities and individual support for all children" could be divided into two sentences.

Language and Grammar: The sentence, "Video data was transcripted," should be corrected to "Video data was transcribed."

INTRODUCTION

The introduction of the scientific article provides a comprehensive background to the study, focusing on the importance of play in Finnish inclusive early childhood education and care (ECEC). It outlines the context, principles, and values of inclusive education in Finland and the significance of pedagogical play.

Here are some strengths and suggestions for improvement:

Strengths:

Clear Context Setting: The introduction effectively sets the context for the study, explaining the role of play in Finnish ECEC and the principles of inclusive education. It provides a solid foundation for readers to understand the importance of the research.

Citation of Relevant Research: The introduction references prior research to support its claims, demonstrating that it is built on a foundation of existing knowledge in the field.

Clarity of Objectives: The introduction clearly outlines the objectives of the study, such as exploring elements of inclusive play and answering research questions related to practical inclusion and shared meaning-making.

Use of Subheadings: The use of subheadings within the introduction aids in structuring the content, making it easier to follow.

Suggestions for Improvement:

Complex Sentences: Some sentences in the introduction are quite long and complex, which may make them harder to follow. Consider breaking them down into shorter sentences to enhance readability.

Specific Examples: While the introduction mentions the importance of inclusive play and pedagogy, providing specific examples or scenarios could help readers better understand the concepts in practice.

Clarification of Acronyms: The acronym "ECEC" is used multiple times without a full explanation of what it stands for. It should be spelled out the first time it's used, followed by the abbreviation in parentheses, and then the abbreviation can be used consistently throughout the article.

Structure: Although the introduction provides valuable information, it could benefit from a brief outline of the structure of the paper, indicating what will be covered in subsequent sections.

METHODS

The methodology section of the article provides a detailed description of how the research was conducted, including data collection and analysis methods. Overall, it offers a comprehensive understanding of the research process. Here are some strengths and suggestions for improvement:

Strengths:

Clear Description of Data Collection: The section provides a clear and detailed explanation of how the data was collected, including the use of video observations, research diaries, and non-participating observations. This clarity helps readers understand the research process.

Justification for Case Study Approach: The rationale for using a case study approach is well-explained, emphasizing its suitability for exploring complex phenomena in real-world settings. This justification provides a solid basis for the research design.

Ethical Considerations: The section addresses ethical concerns related to observing and recording children's activities. It highlights the importance of obtaining informed consent and ensuring the well-being of participants, which is crucial in research involving children.

Data Analysis Method: The description of the data analysis method, particularly the use of narrative analysis, content analysis, and researcher triangulation, is clear and informative. It explains how researchers aimed to respect the meanings and intentions of the participants.

Transparency and Anonymization: The section mentions that efforts were made to ensure transparency in data collection and analysis, such as recording keywords and anonymizing participants' faces. This demonstrates a commitment to ethical research practices.

Suggestions for Improvement:

Clarity in Data Collection Details: While the section provides an overview of data collection, adding more specific details about the types of observations made, the duration of observations, and any specific criteria for selecting participants could enhance understanding.

Explanation of Coding Process: The section mentions the coding process for video clips but does not provide a detailed explanation of the coding categories or criteria used. Providing examples of codes and their meanings would clarify this aspect.

Clarification of Abductive Approach: The use of the term "abduction" in the context of data analysis might be unfamiliar to some readers. It would be helpful to briefly explain what is meant by the "abductive approach" and how it was applied in the analysis.

Alignment with Research Questions: It would be beneficial to explicitly connect the data collection and analysis methods to the research questions stated earlier in the paper. How did these methods help answer the research questions?

Reference Citations: Some claims and statements in this section reference prior research (e.g., Frankenberg research group) without citing the source. Including proper citations for such references is essential for academic integrity.

RESULTS

The "Findings and Discussion" section of the article presents an analysis of three play events in different child groups with diverse linguistic backgrounds. The authors identify and discuss five elements of inclusive play: teachers' participation and active presence in play, enough repetition but a flexible plan and adaptive goals, playful language and the joy of play, enabling tools, non-verbal, and kinesthetic communication, and emerging play, interaction, and long-lasting intensity. Here's a critical evaluation of this section:

Clarity and Organization: The section is relatively well-organized, with clear headings and subheadings. The presentation of findings in narrative form is engaging and helps readers understand the context of each play event.

Detailed Analysis: The authors provide a detailed analysis of each play event, which allows readers to grasp the nuances of inclusive play. The use of direct quotes and descriptions of children's actions enriches the analysis.

Relevance to Research Questions: The section effectively addresses the research questions related to inclusive play and the role of teachers in facilitating it. The identified elements of inclusive play are logically derived from the data presented.

Integration of Literature: The authors make references to existing literature to support their findings and arguments. However, it would be beneficial to provide more in-depth discussions of how their findings align with or contribute to existing research in the field of inclusive education.

Clarity in Language Codes: The use of language codes such as [Finnish], [English], and [other] to explain linguistic variations is a good practice. However, it might be helpful to provide a brief legend or key at the beginning of the section to ensure clarity for readers.

Discussion Depth: While the section presents the identified elements of inclusive play effectively, it could benefit from a deeper discussion of the implications of these findings. How can these elements be practically applied in early childhood education settings? What are the potential challenges in implementing these strategies?

Recommendations: The section mentions the need for recommendations to support inclusive play in ECEC but does not provide specific recommendations. Including actionable suggestions for teachers and policymakers would enhance the practical utility of the research.

Visual Aids: Given the focus on play events, the section might benefit from the inclusion of visual aids, such as diagrams or images from the observations, to help readers visualize the described situations.

DISCUSSION

The "Discussion" section of the article discusses the significance of inclusive play in early childhood education (ECEC) and its role in promoting inclusion, while also presenting recommendations based on the study's findings. Here's a critical evaluation of this section:

Clarity of Argument: The section effectively communicates the importance of inclusive play and its potential benefits in ECEC. It emphasizes the role of play in transcending cultural, linguistic, and physical barriers, which aligns with the theme of inclusion.

Integration of Literature: The section makes appropriate references to existing literature, which supports the argument. However, it could benefit from deeper engagement with the literature to provide a more comprehensive theoretical framework for the discussion.

Recommendations: The section mentions the provision of recommendations based on the study's findings. However, these recommendations are not presented explicitly in this section. To enhance clarity, the authors should consider providing a concise list of actionable recommendations for educators, policymakers, or researchers in the field.

Exploration of Play: The section acknowledges the diverse nature of play, highlighting its various characteristics. This exploration of play is insightful and adds depth to the discussion. However, it could be further enriched by providing concrete examples or case studies illustrating how different types of play can facilitate inclusion.

Multimodal Approach: The mention of play as a "multimodal experience" is valuable. To enhance this point, the authors could provide specific examples of how various modes of communication, such as verbal and non-verbal, are integrated into play to support inclusion.

Pedagogical Significance: The section correctly emphasizes the pedagogical significance of play in ECEC. It aligns with the broader recognition of play as a fundamental aspect of early childhood education. However, to strengthen this argument, the authors could cite additional research that underscores the educational benefits of play.

Cultural Relevance: Given that the study focuses on the Nordic context, it would be beneficial to discuss how the findings and recommendations may have relevance and applicability in different cultural and educational settings. Recognizing the potential cross-cultural applicability of inclusive play strategies would enhance the article's broader impact.

CONCLUSIONS

Clarity of Conclusions: The conclusions drawn in this section are clear and concise. The authors effectively summarize the main findings of the study, emphasizing the importance of participation, inclusion, and the role of play in early childhood education.

Play as a Pathway: The final sentence effectively encapsulates the main message of the article, suggesting that play can serve as a pathway between individual and broader political approaches to inclusion. This provides a thought-provoking conclusion to the article.

Recommendations for Improvement: The section could be enhanced by providing specific examples or case studies that illustrate how play has been successfully used to promote inclusion in different ECEC contexts. This would make the conclusions more tangible and applicable to practitioners.

Global Relevance: While the study primarily focuses on the Finnish context, it would be beneficial to acknowledge the potential relevance of the conclusions to a broader international audience. Recognizing the transferability of inclusive pedagogical practices through play could broaden the impact of the research.

Author Response

Thank you for the revisions, the encouraging feedback, and the important insights on our manuscript! As researchers, we were happy to read about the strengths of our study - a sincere thank you!  Here are the detailed answers for the suggested corrections. We hope we have succeeded in developing the paper. 

Round 2

Reviewer 2 Report

The authors have taken into account the improvement suggestions made. The current manuscript has incorporated the changes that contribute to its quality improvement and can be published. Good job!